# An ImmunoFET Coupled with an Immunomagnetic Preconcentration Technique for the Sensitive EIS Detection of HF Biomarkers

**DOI:** 10.3390/mi15030296

**Published:** 2024-02-21

**Authors:** Hamdi Ben Halima, Nadia Zine, Imad Abrao Nemeir, Norman Pfeiffer, Albert Heuberger, Joan Bausells, Abdelhamid Elaissari, Nicole Jaffrezic-Renault, Abdelhamid Errachid

**Affiliations:** 1Institut de Sciences Analytiques (ISA)-UMR 5280, Université Claude Bernard Lyon 1, 5 Rue de la Doua, 69100 Lyon, France; nadia.zine@univ-lyon1.fr (N.Z.); imad-nmeir@hotmail.com (I.A.N.); abdelhamid.elaissari@univ-lyon1.fr (A.E.); abdelhamid.errachid-el-salhi@univ-lyon1.fr (A.E.); 2Faculty of Arts and Sciences, Holy Spirit University of Kaslik (USEK), Jounieh P.O. Box 446, Lebanon; 3Fraunhofer IIS, Fraunhofer Institute for Integrated Circuits, 91058 Erlangen, Germany; norman.pfeiffer@iis.fraunhofer.de; 4Information Technology (LIKE), Friedrich-Alexander-Universität Erlangen-Nürnberg (FAU), 91058 Erlangen, Germany; albert.heuberger@iis.fraunhofer.de; 5Institute of Microelectronics of Barcelona (IMB-CNM, CSIC), Campus UAB, 08193 Bellaterra, Barcelona, Spain; joan.bausells@imb-cnm.csic.es

**Keywords:** heart failure, interleukin-10, tumor necrosis factor-α, electrochemical impedance spectroscopy, saliva analysis

## Abstract

We propose a new strategy using a sandwich approach for the detection of two HF biomarkers: tumor necrosis factor-α (TNF-α) and interleukin-10 (IL-10). For this purpose, magnetic nanoparticles (MNPs) (MNPs@aminodextran) were biofunctionalized with monoclonal antibodies (mAbs) using bis (sulfosuccinimidyl) suberate (BS_3_) as a cross-linker for the pre-concentration of two biomarkers (TNF-α and IL-10). In addition, our ISFETs were biofunctionalized with polyclonal antibodies (pAbs) (TNF-α and IL-10). The biorecognition between pAbs immobilized on the ISFET and the pre-concentrate antigen (Ag) on MNPs was monitored using electrochemical impedance spectroscopy (EIS). Our developed ImmunoFET showed a low detection limit (0.03 pg/mL) toward our target analyte when compared to previously published electrochemical immunosensors. It showed a higher sensitivity than for other HF biomarkers. Finally, the standard addition method was used to determine the unknown concentration in artificial saliva. The results matched with the expected values well.

## 1. Introduction

The World Health Organization (WHO) estimates that heart failure (HF) will affect 26 million people globally, making it one of the foremost causes of death worldwide [1]. The economic cost of HF is large, and the growing numbers and frequent hospitalizations transform this pathology into a huge economic issue for health care systems; for example, in 2012 HF resulted in costs estimated at around USD 31 billion, which amounts to 10% of the entire health expenditure that is devoted to cardiovascular diseases in the United States (US). However, the total costs are expected to increase by 127% between 2012 and 2030, making it an urgent issue that requires addressing [2]. Also, the overall estimated cost for the European Union is currently EUR 192 billion per year [3].

Tumor necrosis factor-alpha (TNF-α) is a pro-inflammatory cytokine with a polypeptide biochemistry. It is produced as a hormone by either activated monocytes or macrophages. Its main function is to control several biological processes, including the activation of host immunity against neoplastic cell growth, the increased expression of antigens related to allergic reactions [4,5], and cachexia development [6] (cachexia arises from persistent and vasopressor-resistant widening of blood vessels, driven by elevated nitric oxide production that amplifies the presence of the inducible form of nitric oxide synthase, resulting in a states of shock) [7], in addition to its activation role in a multitude of inflammatory processes [8]. 

In healthy subjects, salivary levels of TNF-α can range from a few units to several dozen [9], whereas HF patients are expected to have greatly increased concentrations up to hundreds of units [7,10,11,12]. Clinical and experimental evidence that proves TNF-α’s effects in HF patients continues to accrue. It is firmly established that patients experiencing the onset of heart failure (HF) exhibit elevated concentrations of TNF-α in their bloodstream. Furthermore, there is a direct correlation between TNF-α levels and prognosis. The levels of circulating TNF-α are accountable for the reduced expression of myocardial TNF-α receptors observed in heart failure [13]. Patients with chronic HF exhibit high circulating levels of TNF-α, which are associated with increased severity of their disease [7]. Numerous studies have also confirmed that TNF-α concentration in saliva correlates with its concentration in serum [14], making TNF-α another ideal HF-related salivary biomarker. Moreover, TNF-α concentrations are greater in whole saliva when compared to parotid saliva [15], thus requiring a non-selective sampling of the oral fluid.

Unlike TNF-α, human cytokine synthesis inhibitory factor (CSIF), more commonly known as interleukin-10 (IL-10), plays an anti-inflammatory role and consequently is considered by many to have a protective role in HF [16,17]. IL-10’s biochemistry is that of a 37 kDa homodimer with each monomer made up of 18.5 kDa, 160-amino-acid chains [18]. The cytokine is formed by various cell types, in particular inflammatory cells such as macrophages and T lymphocytes. It functions as the main inhibitor for cytokine synthesis, macrophage activity [16], and the extracellular matrix metalloproteinases [16,19]. IL-10 is found both as a membrane-bound and as a soluble protein [20]. It also prevents the formation of reactive oxygen intermediates [21] and increases soluble TNF-α receptor release [22], which counteracts the actions of TNF-α [23,24,25,26]. IL-10 is upregulated in inflammatory processes as an immune modulator [27,28,29]. Based on these indications, IL-10 is reported to downregulate pro-inflammatory cytokine production, such as IL-1, IL-6, and TNF-α in numerous cell types, and correlate with HF severity [30,31,32]. Administration of IL-10 externally protects against acute lung injury that is caused by oxidative stress, mediated by TNF-α [33], which is amplified by the IL-10 antibody [34]. IL-10 has been documented to alleviate the other negative effects of TNF-α [21]. Additionally, IL-10 is known to alter the protease–anti-protease balance in favor of matrix preservation, thus stimulating the healing of injured myocardium [35]. The regulation of its receptor plays a crucial role in its effects on the tissues [36,37] and the expression of these receptors is changed in patients with HF. IL-10 is usually determined in blood, urine, and saliva [38,39,40,41,42,43,44,45,46,47]. The expected levels for IL-10 are from a few units to a few tens of pg/mL [9].

Various techniques were employed to quantify TNF-α/IL-10 and other biomarkers with the aim of predicting the initial indicators of inflammation [48,49]. These include enzyme-linked immunosorbent assays (ELISAs) [50,51], bioassays [52], radio-immunoassays (RIAs) [53], surface plasmon resonance [54], and other methods [55]. However, the extensive use of these techniques is limited mainly by the requirement for sophisticated technical skills, high instrumentation cost, long run-time, and the impossibility to perform a real-time measurement. Electrochemical biosensors can deliver a timely and reliable medical diagnosis, with benefits including real-time detection, transportability, low-cost operation, and ease of use [56,57]. There have been more than one hundred published papers on electrochemical affinity sensors for the detection of these cytokines and two main review papers [58,59]. Two types of electrochemical immunosensors have been designed: label-free immunosensors or sandwich-type immunosensors. Nanomaterials such as carbon nanotubes, graphene, fullerene, quantum dots, and metallic nanoparticles were used for the amplification of the voltametric signal of the modified electrodes. The obtained detection limits were in the range of pg/mL. Another method of amplification is the use of antibody-functionalized magnetic nanoparticles that are magnetically captured on the surface of the electrode (SPCE) for the detection of TNF-α through an amperometric immunosandwich assay with an LOD of 2 pg/mL [60]. A graphene conductive polymer paper-based sensor was recently developed for the impedimetric detection of TNF-α with an LOD of 5.97 pg/mL. Specifically, biologically sensitive field-effect transistors (BioFETs) represent a highly prevalent category of electronic sensors for biomolecular detection, and their potential in biomedical applications has been extensively demonstrated [61,62]. Detection limits of 1 pg/mL and 5 pg/mL were obtained in Ref. [63] and in Ref. [64], respectively.

In this work, a novel and appealing sandwich method is presented for detecting TNF-α and IL-10. For this purpose, a silicon-nitride-based ISFET was biofunctionalized by immobilizing polyclonal anti-TNF-α/anti-IL-10 antibody on its surface after activation with 11-triethoxysilyl undecanal (TESUD) by a vapor-phase method in a saturated medium using the nucleophilic substitution reaction between the aldehyde and the N-terminus of the antibodies [47,56,63,64,65,66,67]. Meanwhile, magnetic nanoparticles (MNPs) (MNPs@aminodextran) were biofunctionalized with monoclonal antibodies (mAbs) using BS3 as a cross-linker to pre-concentrate two biomarkers (TNF-α and IL-10). Analyte detection was performed through electrochemical impedance spectroscopy (EIS) due to its capacity to discern variations in resistance and capacitance when antibody–antigen recognition occurs, thereby enhancing device sensitivity. The quantification of TNF-α/IL-10 concentration in artificial saliva samples was achieved using the standard addition method (SAM). The results obtained validate our biosensor as a promising tool for TNF-α/IL-10 detection in saliva. To our knowledge, this represents the first silicon-nitride-based ISFET utilizing this strategy for TNF-α/IL-10 detection.

## 2. Materials and Methods

### 2.1. Materials and Chemicals

The wire-bonding process was carried out utilizing aluminum wire (Ø 25 µm) with the Kulicke&Soffa (Singapore) 4523A digital instrument. The ISFET underwent activation using UV/Ozone ProcleanerTM (BioForce, Konstanz, Germany) to create -OH groups on its surface. All experiments were conducted within a Faraday cage at room temperature (20 ± 2 °C). BVT Technologies (Strážek, Czech Republic) provided the counter platinum electrode and the reference Ag/AgCl electrode. EIS measurements were performed using a VMP3 multichannel potentiostat from Biologic-EC-Lab (Seyssinet-Pariset, France), and data acquisition and modeling were implemented with EC-Lab software (V11.30, BioLogic, Seyssinet-Pariset, France). To investigate the surface morphology of the MNPs fixed onto the ISFET, scanning electron microscopy (SEM) was employed with an FEI Quanta FEG 250. BioTechne (R&D Systems, Noyal-Châtillon-sur-Seiche, France) supplied recombinant human IL-10 (217-IL-005), human monoclonal anti-IL-10 antibody (MAB217-SP), human polyclonal anti-IL-10 antibody (AF-217-SP), recombinant human TNF-α (210-TA-005/CF), human monoclonal anti-TNF-α (MAB610-SP), human/mouse polyclonal anti-TNF-α (AF-410-SP), sterile phosphate buffer saline solution (PBS), and PBS containing 0.1% bovine serum albumin (97063-660). HyTest (Turku, Finland) provided recombinant human NT-proBNP (8NT2). Various chemicals including urea (57-13-6), mucin from porcine stomach (84082-64-4), sodium phosphate dibasic (Na_2_HPO_4_) (7558-79-4), anhydrous calcium chloride (CaCl_2_) (10043-52-4), potassium chloride (KCl) (7447-40-7), sodium chloride (NaCl) (7647-14-5), sodium hydroxide (NaOH) (1310-73-2), magnesium nitrate (Mg(NO_3_)_2_) (13446-18-9), sodium bicarbonate (NaHCO_3_) (144-55-8), phosphate buffer saline solution (PBS) tablets (MFCD00131855), ethanolamine (purity ≥ 98%), and pure ethanol (purity 95.0%) (64-17-5) were purchased from Sigma-Aldrich (Saint-Quentin-Fallavier, France). 11-triethoxysilyl undecanal (TESUD, 90%) (116047-42-8) was purchased from ABCR (Karlsruhe, Germany). Bis (sulfosuccinimidyl) suberate (BS3) (82436-77-9) was purchased from Thermo Fisher Scientific (Waltham, MA, USA). Ultrapure water (resistivity > 18 MΩ cm) was produced by the Elga PURELAB Classic system (Veolia, Aubervière, France). PBS tablets were used to create PBS buffer by dissolving it in ultrapure water, thus yielding a 0.01 M phosphate buffer (pH 7.4) with 0.0027 M potassium chloride and 0.137 M sodium chloride as indicated by the supplier. EPO TEK H70E2LC epoxy resin (Parts A and B) was supplied from Epoxy Technology Inc. (Paris, France).

### 2.2. ISFET Device Fabrication

Ion-sensitive field effect transistor (ISFET) devices featuring a 100 nm thick silicon nitride gate as a dielectric layer were constructed on fully depleted p-type <100> silicon-on-insulator (SiO_2_) substrates. These substrates were polished 4-inch wafers with a resistivity of 4–40 Ω/cm. An 800 nm silicon oxide insulator was thermally grown (Figure 1a) in a wet oxidation process at 1100 °C. The initial standard photolithographic process delineated the transistor’s drain and source areas by wet-etching the exposed silicon oxide layer (Figure 1b). Subsequently, these areas were doped via ion implantation with phosphorus atoms (Figure 1c) at a dose of 4.2 × 1015 at/cm^2^ @ 100 keV, defining controlled n-type regions with a specific number of free charges.

The second photolithography step in the photolithographic process determined the gate of the ISFET and the bulk’s window contact (Figure 1d), later forming a 100 nm thick silicon nitride layer through low-pressure chemical vapor deposition (LPCVD) over a thermally grown 78 nm thick silicon oxide layer (Figure 1e). A double photolithographic process with hard-baked resist (Figure 1f,g) defined the gate’s structure through an anisotropic reactive ion-etching (RIE) process.

Next was the deposition and insulation of a lift-off photolithographic resist (Figure 1h), defining metallic contacts and traces from electric pads to the ISFET’s source and drain. For metallic contacts and electric pads, a 150 nm conductive layer of platinum over a 15 nm layer of titanium, acting as a diffusion barrier, was deposited through a physical vapor deposition (PVD) sputtering process (Figure 1i). Finally, the metallic bilayer was defined by lifting off the resist (Figure 1j) with an ultrasonic bath of acetone.

The final step in the planar process technology for ISFET fabrication involved protecting the device’s surface from environmental damage using a double passivation layer that covers the entire surface, except for the open active areas like the 20 µm × 400 µm sensitive gate ISFET area, the reference electrode, and the electric pads (Figure 1k). This passivation layer consisted of 400 nm of Si_3_N_4_ over 400 nm of SiO_2_, deposited using a plasma-enhanced chemical vapor deposition (PECVD) method.

To use the ISFET devices in liquid solutions, the silicon wafer was cut into pieces, and the resulting chips were pasted onto a printed circuit board (PCB) using EPO TEK H70E2LC epoxy resin from Epoxy Technology (France). Wire-bonding was performed with Kulicke&Soffa 4523 (Figure 2), using 25 µm diameter aluminum wires. Subsequently, the bonding wires, the external areas of the chip, and the electric tracks on the PCB were enclosed in the same epoxy resin to protect them from exposure during the experiments (Figure 3). Lastly, the reference electrode (RE) based on Ag/AgCl was produced through electrochemical deposition of Ag on a platinum microelectrode, utilizing sodium nitrate (NaNO_3_) (1 M) and silver nitrate (AgNO_3_) (25 mM) at pH 1. The deposited silver layer underwent chlorination through overnight incubation of the RE in a HCl (1 M) solution. A comprehensive description of the RE fabrication process can be found in prior work by our group [68,69]. As a standard procedure, we perform an electrical characterization of the devices when a batch is fabricated, and the typical values that we measure for the leakage currents are on the order of IG = 10–100 pA for the gate current and ID = 1–10 nA for the drain current.

### 2.3. Synthesis and Characterization of MNPs

#### 2.3.1. Seed Magnetic Latex Particle Preparation

An amphiphilic polymer (polyacrylic acid containing hydrophobic groups) solution with a concentration equal to 0.5 g/L and pH = 9 was used as a stabilizing agent to wash the oil-in-water magnetic emulsion three times. Deionized water was employed for the last emulsion wash under nitrogen. A 2 g dispersion of the stabilized magnetic emulsion was placed in a 50 mL glass reactor under a nitrogen stream and stirring for 30 min before adding divinyl benzene (DVB) (900 mg) then stirring continued for 1 h before introducing KPS initiator (potassium persulfate) (18 mg) dissolved in 1 mL of deionized water. The polymerization reaction was carried out at 70 °C for 20 h, and the polymerization conversion was determined and found to be 95%.

#### 2.3.2. Preparation of Amine-Containing Dextran Polymer

A mass of 50 g of DextranT40 was dissolved in 250 mL of water before adding NaIO_4_ (26.4 g) and 1.6-hexamethylenediamine (31.5 g). NaIO_4_ was employed to obtain an oxidized dextran solution. The mixture yielded a homogeneous orange solution to which a solution of sodium borohydride (18.6 g) in 200 mL of 1 mM aqueous potassium hydroxide was added. The result was a yellow–orange solution. After freeze-drying of the amino dextran solution over 48 h, 16 g of pale-yellow crystals was obtained, which corresponded to a yield of 32%.

#### 2.3.3. Adsorption of Amino Dextran onto Magnetic Latex Particles

The prepared seed magnetic latex particles (20 mL at 4 wt.%) were washed thrice using a solution of Triton X-100 at a 1 g/L concentration before adding 50 mL of amino dextran solution (13 g/L). The adsorption was carried out under stirring overnight. Finally, 10 mM NaCl solution was used to wash the functionalized magnetic latex particles.

Once the synthesis was completed, these MNPs were characterized by different techniques in order to ensure their quality as magnetic particles and that they carried the right chemical function that will be implemented for the immobilization of the antibodies.

#### 2.3.4. Transmission Electron Microscopy Analysis

Figure 3 represents a transmission electron microscopy image of the functionalized magnetic latex particles. As clearly shown, the magnetic core is surrounded by a polymer shell. This shell mainly results from the polymerization step, whereas the adsorbed amino dextran’s thickness cannot be seen on the surface of the particles.

#### 2.3.5. Hydrodynamic Particle Size

The hydrodynamic size (Dh) of the prepared functional magnetic latex particles was measured using dynamic light scattering, and the obtained size distribution is shown in Figure 4. The prepared magnetic latex particles are submicron in size, with narrow size distribution, and the average hydrodynamic size was found to be 302 nm. The observed narrow size distribution reflects the absence of aggregated particles.

### 2.4. Standard Solutions

Human anti-IL-10 monoclonal antibody (mAb), human anti-IL-10 polyclonal antibody (pAb), human anti-TNF-α monoclonal antibody, and human anti-TNF-α polyclonal antibody were prepared using the supplier’s procedure at 0.5 g L^−1^ in RB01 buffer. Subsequently, they were diluted using the standard solution to 10 µg mL^−1^ and they were kept at −20 °C until use. Their corresponding antigens (IL-10 and TNF-α) were similarly prepared at 0.1 g L^−1^ in the RB02 buffer.

NT-proBNP antigen was diluted with PBS 1× and stored at −80 °C until use. Before use, 10 μL of each antigen solution was carefully defrosted at 4 °C for 15 min before further diluting it in PBS 1× to standard working solutions.

### 2.5. Preparation of Artificial Saliva (AS)

Artificial saliva (AS) was prepared in accordance with the following literature [64]. First, 0.6 mg/mL of sodium phosphate dibasic (Na_2_HPO_4_), 0.4 mg/mL of potassium chloride (KCl), 4 mg/mL of urea, 0.6 mg/mL of anhydrous calcium chloride (CaCl_2_), 0.3 mg/mL of sodium bicarbonate (NaHCO_3_), 4 mg/mL of mucin from porcine stomach, and 0.4 mg/mL of sodium chloride (NaCl) were dissolved together using an ultrasonic bath for 20 min until homogeneity in Millipore Milli-Q nanopure water (resistivity > 18 MΩ cm) which was produced by a Millipore reagent water system. Upon complete dissolution, the pH was readjusted with 0.1 M of sodium hydroxide (NaOH) to the value of 7.2. The prepared AS was stored at 4 °C until further use.

### 2.6. Sample Preparation

TNF-α and IL-10 were quantified in AS by performing the standard addition method. A constant volume of AS (5 μL) was added to each of four 1.5 mL Eppendorf tubes. The first concentration was fixed to a final volume of 1 mL with 950 μL of PBS 1× only (Aliquot 1). The remaining concentrations had an additional 30, 60, and 90 μL of a 100 g/mL TNF-α/IL-10 standard solution incorporated in them, respectively. Finally, each concentration was then completed to 1 mL with PBS 1× (Aliquots 2, 3, and 4) [63,65,66]. Every standard solution and sample underwent analysis in quadruplicate (n = 4). EIS analyses were conducted following the description provided in the subsequent subsection.

### 2.7. Biofunctionalization of MNPs

The general process for MNP biofunctionalization requires that 10 µL of a mother solution containing MNPs@aminodextran (size about 300 nm, 0.02% solid content) [70] be rinsed with 1 mL of 10 mM PBS 1× (pH 7.4). MNP immobilization was carried out by using a magnetic rack placed beside the tube. The rinsing was repeated three times to ensure no residue remained. Amine groups present on the MNPs were activated by incubating MNPs in 200 µL of bis (sulfosuccinimidyl) suberate (BS_3_) (cross-linker) (10 µg/mL) at room temperature (20 ± 2 °C) under soft stirring (33 rpm) for 10 min. The residual BS_3_ was then removed, and the MNPs were washed twice using 200 µL of PBS 1×. Then, 100 µL of mAb (10 µg/mL) was added, and the mixture was slowly stirred at room temperature for 3 h, until mAb-MNPs@aminodextran (mAb-TNF-α) and mAb-IL-10 complexes were finally obtained. The MNPs were immobilized again by a magnetic rack, and the solvent was removed. Afterwards, 500 µL of 1% ethanolamine in PBS 1× was added to the tube to deactivate the unreacted sites on the MNPs. The mixture was stirred (33 rpm) at room temperature for 30 min. After that, the unreacted ethanolamine was removed, and the complex was washed two times with 1 mL of 10 mM PBS 1×. A magnetic rack was used to immobilize the nanoparticles and remove only the supernatants. Then, Ag was pre-concentrated by incubating the functionalized MNPs in Ag standard solution for 30 min to allow the formation of the complex with the MNPs@aminodextran-antibody (Ag-Ab-MNPs). Different Ag concentrations (5, 10, 20 pg/mL) were tested for both TNF-α and IL-10. Then, EIS measurements were carried out to characterize the ImmunoFETs.

### 2.8. Biofunctionalization of the ImmunoFET

The biofunctionalization procedure was the same for each biomarker and included the following steps: the chip was initially cleaned by sonication in acetone, then thoroughly rinsed with Milli-Q water. The device was then irradiated with UV/Ozone Procleaner^TM^ (BioForce, Germany) to activate its surface by creating active hydroxyl groups (-OH) on which to graft the silane aldehydes. Then, the vapor-phase method was used to allow the activated surface to react with (11-triethoxysilyl) undecanal (TESUD). Afterward, the chips were heated in an oven at 100 °C for 1 h and then rinsed with pure ethanol and dried under nitrogen flow. Subsequently, the functionalized devices were incubated in a 10 µg/mL standard solution of the target pAb-TNF-α/pAb-IL-10. Finally, to prevent non-specific binding during the detection, the remaining active aldehyde groups were deactivated using ethanolamine solution (1% in PBS 1×) for 45 min at room temperature (20 ± 2 °C).

### 2.9. Electrochemical Measurements

EIS measurements were made in 7 mL PBS 1× using a frequency window of 10 kHz to 10 Hz at two frequency points per decade and a fixed voltage amplitude of 75 mV (E_ac_) without an applied potential (E_dc_) (0 V) versus the Ag/AgCl reference. EIS spectra were fitted with the Randomize (5000 iterations) + Simplex method (fit stopped at 5000 iterations) with the Randles equivalent circuit model [R_1_ + Q_2_/R_2_]: R_1_ is the electrolyte solution’s resistance (PBS 1×), the parallel elements are Q_2_ (the coefficient of the constant phase element), and R_2_ is the charge transfer resistance (R_ct_). The R_2_ is obtained after measuring each TNF-α/IL-10 antigen concentration with the ISFET. It is then normalized by subtracting it along with the R_2_ of the corresponding antibody from the same ISFET. Then, the result is divided by, again, the same R_2Ab_ of the antibody following this equation: |R_2Ag_ − R_2Ab_|/R_2Ab_ to find the value ∆R/R. Each ∆R/R is then plotted against its corresponding antigen concentration.

## 3. Results

### 3.1. Coupling the Immunomagnetic Pre-concentration Process with the ImmunoFET

The use of MNPs to perform a sandwich assay is a common strategy for the development of biosensors. This method is used to avoid the matrix effect that is a major obstacle for electrochemical biosensors. It also acts as a signal amplifier. However, their use increases the complexity of the biosensor by adding more steps, which ultimately increases its cost.

To confirm the binding between MNPs@aminodextran-antibody (Ag-mAb-MNPs) and the target pAb bound onto the chip, scanning electron microscopy (SEM, FEI Quanta FEG 250 instrument, ELCMI, Zarogoza, Spain) images were taken. An example of the “capturing” of the Ag-Ab-MNPs by pAb is presented in Figure 5 and Figure 6, respectively, for Ab = IL-10 and Ab = TNF-α. The pAb-Ag-mAb-MNP complexes remained on the surface of the electrode even after aggressive rinsing with PBS 1×. These images clearly show that the area of the gate of the ImmunoFET is completely covered by the MNPs; this proves that the interaction between the MNPs covered with the antibodies and antigen reacted in the right way with the pAb immobilized on the gate of the ImmunoFET.

### 3.2. Determination of HF Biomarker in Standard Solutions by ImmunoFET

Standard solutions containing the target biomarker (e.g., TNF-α, IL-10) in increasing concentrations (0.1, 2, 5 pg/mL) were prepared for analysis by EIS to investigate the ImmunoFET’s responsiveness when combined with MNPs functionalized with mAb (TNF-α/IL-10). The ImmunoFET was incubated in each sample for 45 min, and then EIS measurements were performed. By way of illustration, Figure 7A shows Nyquist plots obtained by analyzing IL-10 standard solutions in PBS 1×. Detection of IL-10 from 0.1 pg/mL to 5 pg/mL increased the R_ct_, forming a clear distinction between each observable concentration analyzed and confirming that our ImmunoFETs are highly sensitive to IL-10. Figure 7B shows the sensitivity curves obtained using the ImmunoFET functionalized with mAb-IL-10 by analyzing standard solutions containing possible interfering species, represented by the other target HF biomarkers (e.g., NT-proBNP, TNF-α). A good correlation (R^2^ always > 0.99) between analyte concentration and EIS signals (intended as ∆R/R = (R_2Ag_ − R_2Ab_)/R_2Ab_, with R_2_ = R_ct_) was found for each biomarker. Interference studies were also carried out to confirm the selectivity of the devices. The developed ImmunoFETs were shown to be highly selective toward IL-10 when compared to the other HF biomarkers. The R_ct_ of NT-proBNP (red) (R^2^ = 0.7555 with a slope of 0.0019) and TNF-α (blue) (R^2^ = 0.524 with a slope of 0.0013) was much lower than that IL-10 (R^2^ = 0.991 with a slope of 0.0452). The sensitivity of our ImmunoFETs was twenty-four times higher than that for NT-proBNP and thirty-five times higher than that for TNF-α which proves the high overall performance of the developed ImmunoFET.

The same methodology was used for the characterization of the ImmunoFET for the detection of TNF-α in PBS 1×. The ImmunoFET exhibits a linear increase in electrochemical impedance response upon combining MNPs with the specific antigen TNF-α, in a range of antigen concentrations from 0.1 pg/mL to 5 pg/mL (Figure 8). The obtained ImmunoFET did not exhibit any significant change when subjected to NT-proBNP and IL-10 interferents (Figure 8B), revealing high specificity of the ImmunoFET to TNF-α.

The analytical performance of our developed ImmunoFETs was compared to those of the published electrochemical immunosensors for the detection of TNF-α/IL-10 (Table 1). The detection limit of ImmunoFETs is in the lower range, and its detection range falls in the range of concentrations of TNF-α/IL-10 for a healthy person and for a person with HF. In addition, this system is easy to use, small in size, and has strong potential as a point-of-care testing system.

### 3.3. Determination of HF Biomarkers in Artificial Saliva Using the Standard Addition Method

To simulate the analyte quantification in an “unknown” sample, samples containing the target biomarker (IL-10) in artificial saliva (AS) were also analyzed. An AS sample was spiked with the analyte (1000 pg/mL), and then 5 µL of it was added to four 1.5 mL Eppendorf tubes. To reach a final volume of 1 mL, 950 µL of PBS 1× was added to the first Eppendorf tube (Aliquot 1). IL-10 standard solution in PBS 1× (100 pg/mL) was then added in increasing volumes to the next Eppendorf tube, and each tube was then made up to volume (1 mL) with PBS 1× (Aliquots 2, 3, and 4). The EIS measurement was carried out for each prepared sample. By way of illustration, Figure 9 shows Nyquist plots obtained by analyzing the “unknown” AS sample obtained by performing the standard addition method for detecting IL-10. Figure 9B shows the corresponding calibration curve obtained by performing a linear data fitting. The signal linearly increased with the sample concentration, proving the efficiency of IL-10 detection. This curve was used to extrapolate IL-10 concentration in the “unknown sample.” As can be seen from R^2^ = 0.9951, the experimental points are aligned, the Y-intercept corresponds to IL-10 concentration in the sample without the standard addition (Aliquot 1), and the X-intercept (in absolute value), multiplied by the corresponding dilution factor, represents IL-10 concentration in the “unknown” sample. The obtained concentration (4.83 × 200 = 968 pg/mL) was in good agreement with the expected concentration (1000 pg/mL) [47,60,61,62].

The prepared samples containing the target biomarker (TNF-α) in AS were also analyzed using the same procedure described before. Figure 10 shows the Nyquist plots (Figure 10A) and the corresponding calibration curve (Figure 10B), obtained by performing a linear data fitting. This curve was used to extrapolate TNF-α concentration in the “unknown sample”. As can be seen from R^2^ = 0.9948, the experimental points are aligned, Y-intercept corresponds to TNF-α concentration in the sample without the standard addition (Aliquot 1), and the X-intercept (in absolute value), multiplied by the corresponding dilution factor, represents TNF-α concentration in the “unknown” sample. The obtained concentration (4.95 × 200 = 991 pg/mL) was in good agreement with the expected concentration (1000 pg/mL).

For both experiments, the obtained results for both the “unknown samples” corresponding to IL-10 and TNF-α were in excellent concordance with what was prepared before spiking the AS samples.

## 4. Conclusions

In conclusion, we successfully optimized the protocol for the biofunctionalization of MNPs with monoclonal antibodies with their corresponding antigens. In addition, our ISFET was successfully biofunctionalized with a polyclonal antibody. The interaction between the pAb immobilized onto the ISFET and the antigen pre-concentrate with MNPs was then studied using electrochemical impedance spectroscopy. Our developed ImmunoFET showed good sensitivity and selectivity toward our target analyte with a low detection limit (0.03 pg/mL) when compared to other HF biomarkers. Finally, the standard addition method was used in order to determine the unknown concentration in artificial saliva. The results showed a good agreement with the expected values. The global analytical system will be integrated into a microfluidic point-of-care microsystem, including both stages: immunomagnetic preconcentration and EIS detection on the immunoFET.

## Figures and Tables

**Figure 1 micromachines-15-00296-f001:**
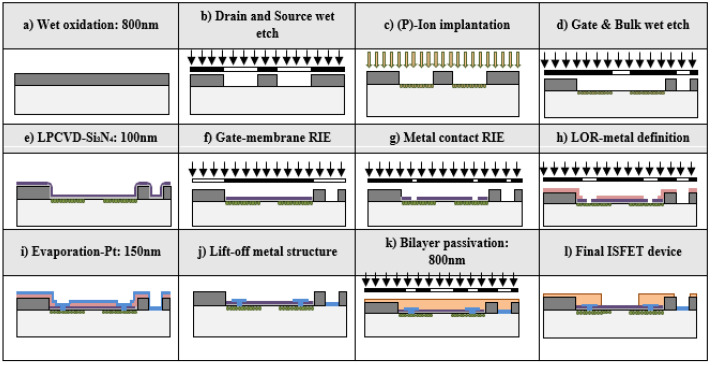
(**a**) The 800 nm SiO_2_ wet oxidation; (**b**) drain and source area wet-etching; (**c**) phosphorus ion implantation for n-type region formation; (**d**) gate and bulk definition through wet-etching; (**e**) 78 nm SiO_2_ thermally grown and 100 nm Si_3_N_4_ with LPCVD as insulator material for ISFET gate; (**f**) 1st of 2 photolithographic processes for gate definition; (**g**) 2nd photolithography and RIE process; (**h**) lift-off resist definition for metallic contacts; (**i**) 150 nm of Pt over 15 nm of Ti PVD process; (**j**) lift-off process for metallic part definition; (**k**) passivation bilayer of 400 nm Si_3_N_4_ over 400 nm SiO_2_ CVD process; (**l**) final ISFET device.

**Figure 2 micromachines-15-00296-f002:**
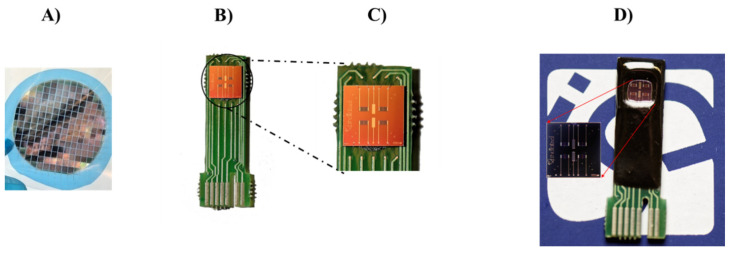
(**A**) Wafer after the fabrication process; (**B**) Chip glued and wire-bonded to the PCB; (**C**) Wire-bonding detail; (**D**) Photograph of the encapsulated chip containing four ISFETs and two Ag/AgCl reference microelectrodes.

**Figure 3 micromachines-15-00296-f003:**
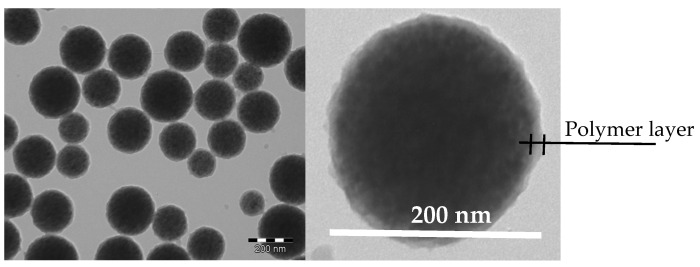
TEM image of the functionalized magnetic latex particles bearing amino dextran shell layer.

**Figure 4 micromachines-15-00296-f004:**
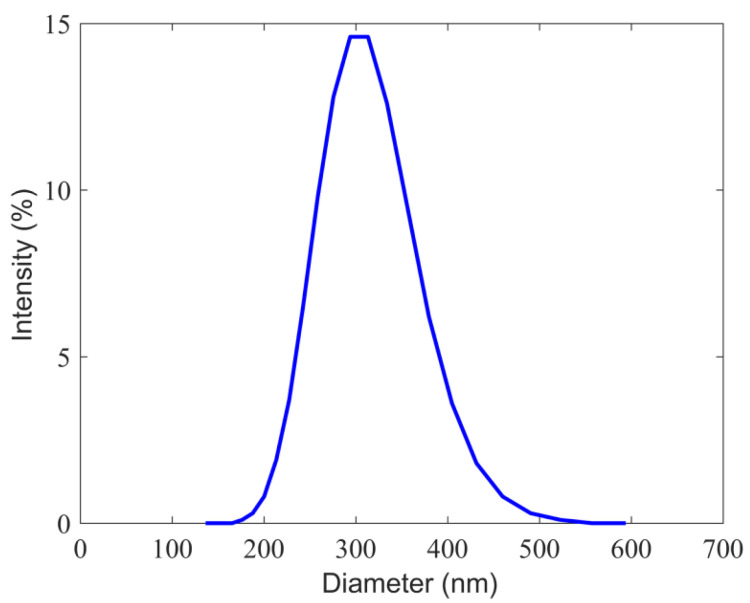
Hydrodynamic size distribution of functional magnetic latex particles measured in 1 mM NaCl solution and at 20 °C.

**Figure 5 micromachines-15-00296-f005:**
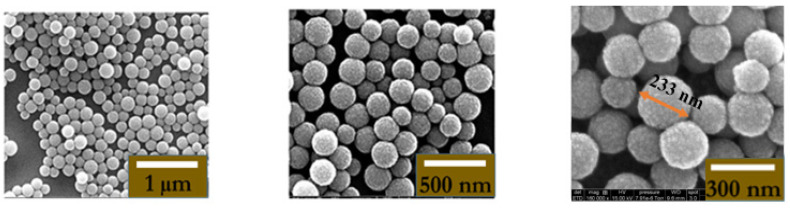
Three different resolutions (1 µm; 500 nm; 300 nm) of SEM (FEI Quanta FEG 250 instrument) image of Ag-Ab-IL-10-MNPs@aminodextran bonded to p-Ab present on the ImmunoFET.

**Figure 6 micromachines-15-00296-f006:**
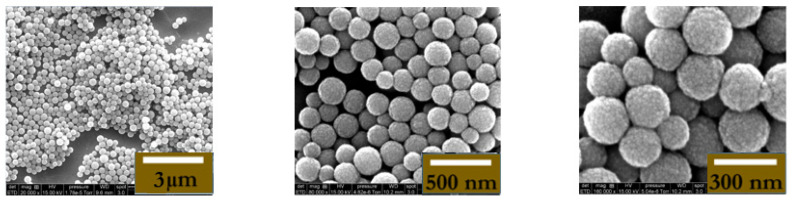
Three different resolutions (3 µm; 500 nm; 300 nm) of SEM (FEI Quanta FEG 250 instrument) image of Ag-Ab-TNF-α-MNPs@aminodextran bonded to pAb present on the ImmunoFET.

**Figure 7 micromachines-15-00296-f007:**
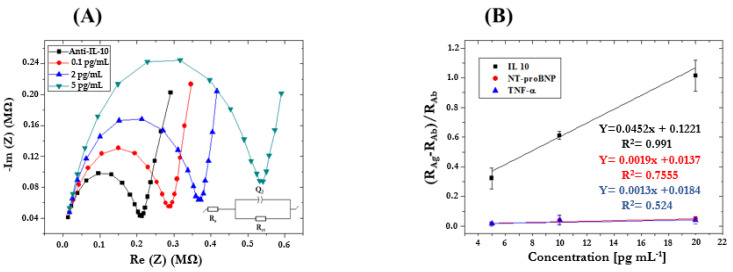
(**A**) Nyquist plots obtained by analyzing IL-10 standard solutions in PBS 1× (concentration range 0.1–5 pg/mL; Frequency range 10 kHz–10 Hz; E_ac_ 75 mV; and E_dc_ 0 V); (**B**) Sensitivity curves obtained using ImmunoFET functionalized with pAb-IL-10 and by analyzing standard solutions containing interfering species (e.g., NT-proBNP, TNF-α) in the same concentration range after repeating it 3 times.

**Figure 8 micromachines-15-00296-f008:**
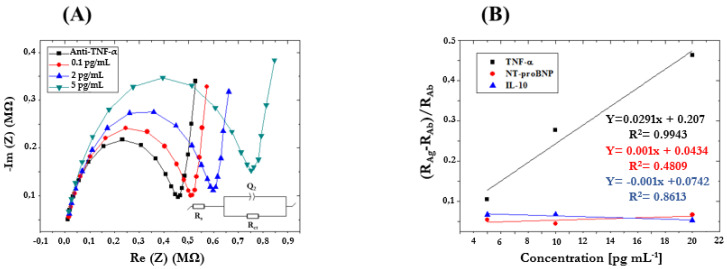
(**A**) Nyquist plots obtained by analyzing TNF-α standard solutions in PBS 1× (concentration range 0.1–5 pg/mL; Frequency range 10 kHz–10 Hz; Eac 75 mV; and Edc 0 V); (**B**) Sensitivity curves obtained using ImmunoFET functionalized with pAb-IL-10 and by analyzing standard solutions containing interferences (e.g., NT-proBNP, TNF-α) in the same concentration range.

**Figure 9 micromachines-15-00296-f009:**
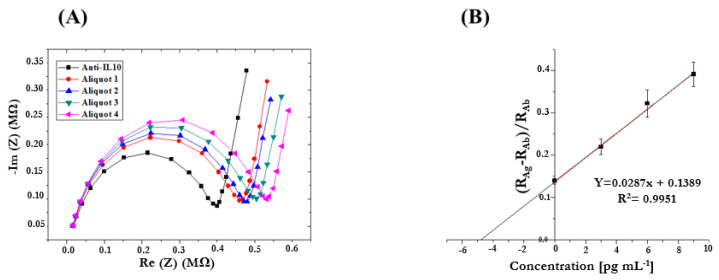
(**A**) Nyquist plots obtained by analyzing IL-10 in AS samples by performing the standard addition method (Frequency range 10 kHz–10 Hz, Eac 75 mV, and Edc 0 V); (**B**) Calibration curve obtained by performing the standard addition method to detect IL-10 in AS samples.

**Figure 10 micromachines-15-00296-f010:**
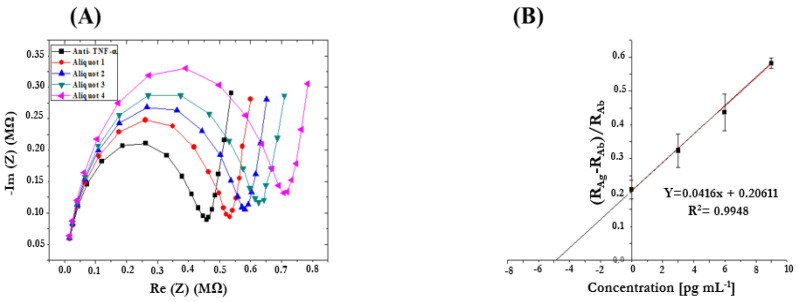
(**A**) Nyquist plots obtained by analyzing TNF-α in AS samples by performing the standard addition method (Frequency range 10 kHz–10 Hz, Eac 75 mV, and Edc 0 V); (**B**) Calibration curve obtained by performing the standard addition method to detect TNF-α in AS samples.

**Table 1 micromachines-15-00296-t001:** Comparison of different electrochemical immunosensors for TNF-α/IL-10 detection.

Technique	Electrode	Immobilizing Biomolecules	Analyte	Linear Range	LOD	Reference
EIS	Substrate “HfO_2_”	mAb-IL-10	PBS 1×	0.1 pg/mL–50 ng/mL	-	[57]
EIS	Au	mAb-IL-10	PBS 1×	1–15 pg/mL	0.3 pg/mL	[71]
EIS	Substrate “Si_3_N_4_”	mAb-IL-10	PBS 1×	0.1–50 pg/mL	0.3 pg/mL	[72]
EIS	ISFET “Si_3_N_4_”	pAb-IL-10	AS	0.1–5 pg/mL	0.03 pg/mL	This work
EIS	Au	mAb-TNF-α	AS	1–15 pg/mL	1 pg/mL	[73]
Mott–Schottky	Substrate “Si_3_N_4_”	mAb-TNF-α	AS	1–30 pg/mL	1 pg/mL	[74]
Amperometry	Au	mAb-TNF-α	AS	1–15 pg/mL	0.3 pg/mL	[75]
Electrical measurement	ISFET “Si_3_N_4_”	mAb-TNF-α	AS	5–20 pg/mL	5 pg/mL	[64]
EIS	ISFET “Si_3_N_4_”	pAb-TNF-α	AS	0.1–5 pg/mL	0.03 pg/mL	This work

## Data Availability

The data that support the findings of this study are available from the corresponding author upon reasonable request.

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
