# Peer review of "An ImmunoFET Coupled with an Immunomagnetic Preconcentration Technique for the Sensitive EIS Detection of HF Biomarkers"

_micromachines, 2024, doi:10.3390/mi15030296_

Round 1
Reviewer 1 Report
Comments and Suggestions for Authors
It’s a well-prepared work and can be considered for further publication upon a few modifications.
1. Mention the concentration of PBS wherever been used.
2. Measure the leakage current.
3. Update the scale bar in Figure 6.
4. Core-shell structure is not visible from the TEM (figure 3). Please correct it.
5. Please provide the references for the calculation of the concentration of the unknown solution. How do you get the dilution factor of 200?
Author Response
Reviewer #1
Title: An immunoFET coupled with an immunomagnetic preconcentration technique for the sensitive EIS detection of HF biomarkers.
It’s a well-prepared work and can be considered for further publication upon a few modifications.
- Mention the concentration of PBS wherever been used.
The PBS used has been described in the materials and methods as a tablet, as such, whenever it is mentioned, we added the quantity of the tablet used for the preparation. Coincidentally, it is always 1X.
- Measure the leakage current.
We are usually working with these ISFET devices. As a standard procedure, we perform an electrical characterization of the devices when a batch is fabricated, and the typical values that we measure for the leakage currents are in the order of IG= 10-100 pA for the gate current and ID=1-10 nA for the drain current.
- Update the scale bar in Figure 6.
The scale bars have been updated to a more consistent form.
- Core-shell structure is not visible from the TEM (figure 3). Please correct it.
The polymer layer is well visible on Fig. 3B.
- Please provide the references for the calculation of the concentration of the unknown solution. How do you get the dilution factor of 200?
References have been added which explain our methodology. As for the dilution factor of 200, it is obtained by diluting 5 µL of artificial saliva which contains the unknown concentration of TNF-α or IL-10 in 950µL of PBS.
Reviewer 2 Report
Comments and Suggestions for Authors
Title: An immunoFET coupled with an immunomagnetic preconcentration technique for the sensitive EIS detection of HF biomarkers.
The authors proposed a sandwich approach for detection of two HF biomarkers TNF-alpha, and IL-10, with using magnetic nanoparticles biofunctioned with mono-antibodies to pre-concentrate the two biomarkers. EIS was applied to detect pre-concentrated antigens. Low pg/ml detection limit was achieved.
1. The literature survey seems not sufficient. Other studies with EIS and heart failure keywords should be included with a thorough search. There are studies with electrochemical standard sensors, microfluidic and paper-based biosensors for this purpose. They should be included and performances should be compared for different types of chips.
2. Section 2.1, the reagents model number should be included, not only the brand.
3. Section 2.2, is the fabrication approach refer to other studies or self-developed by the authors?
4. Figure 7b: What is the measurement number for each error bar?
5. What is the disadvantages of the developed approach for heart failure diagnostics? A discussion should be added besides its advantages.
6. The key performance statistics should be reported in the conclusions.
7. Any interference studies or selectivity studies for different biomarkers studied?
Comments on the Quality of English LanguageThe quality of the english language is fine.
Author Response
Reviewer #2
Title: An immunoFET coupled with an immunomagnetic preconcentration technique for the sensitive EIS detection of HF biomarkers.
The authors proposed a sandwich approach for detection of two HF biomarkers TNF-alpha, and IL-10, with using magnetic nanoparticles biofunctioned with mono-antibodies to pre-concentrate the two biomarkers. EIS was applied to detect pre-concentrated antigens. Low pg/ml detection limit was achieved.
- The literature survey seems not sufficient. Other studies with EIS and heart failure keywords should be included with a thorough search. There are studies with electrochemical standard sensors, microfluidic and paper-based biosensors for this purpose. They should be included and performances should be compared for different types of chips.
Literature survey was completed as follows (lines103-121):
There have been more than one hundred published papers on electrochemical affinity sensors for the detection of these cytokines and two main review papers [Filik, H.; Avan, A. Electrochemical immunosensors for the detection of cytokine tumor necrosis factor alpha: A review. Talanta 2020, 21, 120758. doi:10.1016/j.talanta.2020.120758. Perez, D.J.; Patiño, E.B.; Orozco, J. Electrochemical nanobiosensors as point-of-care testing solution to cytokines measurement limitations. Electroanalysis 2022, 34, 184-211. doi: 10.1002/elan.202100237]. Two types of electrochemical immunosensors were designed: label-free immunosensors or sandwich-type immunosensors. Nanomaterials such as carbon nanotubes, graphene, fullerene, quantum dots, metallic nanoparticles were used for the amplification of the voltametric signal of the modified electrodes. The obtained detection limits were in the range of pg/mL. Another way of amplification is the use of antibody-functionalized magnetic nanoparticles that are magnetically captured on the surface of the electrode (SPCE) for the detection of TNF-a through an amperometric immunosandwich assay with an LOD of 2 pg/mL [Eletxigerra, U.; Martinez-Perdiguero, J.; Merina, S.; Villalonga, R.; Pingarrón, J.M.; Campuzano, S. Amperometric magnetoimmunoassay for the direct detection of tumor necrosis factor alpha biomarker in human serum. Anal Chim Acta 2014, 838, 37-44. doi: 10.1016/j.aca.2014.05.047]. A graphene conductive polymer paper-based sensor was recently developed for the impedimetric detection of TNF-a with an LOD of 5.97 pg/mL.
- Section 2.1, the reagents model number should be included, not only the brand.
We have added the catalogue and CAS number for all the reagents and proteins used
- Section 2.2, is the fabrication approach refer to other studies or self-developed by the authors?
The approach for fabrication of the ISFET devices is based on a long experience by the coauthors at IMB-CNM, which started in the early 1990’s [1,2]. Refs. [3,4] show some later developments, and refs. [5,6] describe some recent work. “
This fabrication technology is based on the experience by the authors on the development of ISFET devices.
[1] A. Merlos, C. Cané, J. Bausells, J. Esteve, Modelization and fabrication of ISFET based sensors, Microelectronic Engineering 15 (1991) 423-426.
[2] S. Alegret, J. Bartrolí, C. Jiménez-Jorquera, M. del Valle, C. Domínguez, J. Esteve, J. Bausells, Flow-through pH-ISFET + reference-ISE as integrated detector in automated FIA determinations, Sensors and Actuators B 7 (1992) 555-560.
[3] A. Errachid, N. Zine, J. Samitier, J. Bausells, FET-based chemical sensor systems fabricated with standard technologies, Electroanalysis 16 (2004) 1843-1851.
[4] M. Castellarnau, N. Zine, J. Bausells, C. Madrid, A. Juárez, J. Samitier, A. Errachid, Integrated cell positioning and cell-based ISFET biosensors, Sensors and Actuators B 120 (2007) 615–620.
[5] D. Vozgirdaite, H. Ben Halima, F.G. Bellagambi, A. Alcacer, F. Palacio, N. Jaffrezic-Renault, N. Zine, J. Bausells, A. Elaissari, A. Errachid, Development of an ImmunoFET for analysis of Tumour Necrosis Factor-α in artificial saliva: application for heart failure monitoring, Chemosensors 9 (2021) 26.
[6] J. Bausells, H. Ben Halima, F.G. Bellagambi, A. Alcacer, N. Pfeiffer, M. Hangouët, N. Zine, A. Errachid, On the impedance spectroscopy of field-effect biosensors, Electrochemical Science Advances, 2 (2022) e2100138.
- Figure 7b: What is the measurement number for each error bar?
Each error bar was measured three times. We have added this information in the figure’s legend.
- What is the disadvantages of the developed approach for heart failure diagnostics? A discussion should be added besides its advantages.
The advantages and disadvantages of the methods used were further explored in section 3.1.
- The key performance statistics should be reported in the conclusions.
We added the detection limit in the conclusion.
- Any interference studies or selectivity studies for different biomarkers studied?
We did perform interference and selectivity studies for the targeted biomarkers. When we tested for IL-10 we used TNF-α and NT-proBNP. And for TNF-α, we used IL-10 and NT-proBNP.
Round 2
Reviewer 1 Report
Comments and Suggestions for Authors
Accept for the publication
Reviewer 2 Report
Comments and Suggestions for Authors
The authors have addressed the comments from this reviewer.